# OpenReview forum: "Break the Brake, Not the Wheel: Untargeted Jailbreak via Entropy Maximization"
_TMLR — Under review for TMLR_

### Review · Reviewer_gGGt · 2026-06-01

**Summary Of Contributions:**

## Summary
This paper focuses on the transferability of jailbreak attacks on VLMs. It proposes an entropy-based optimization objective to improve transferability and further introduces a KL regularization term to preserve text quality.

## Strengths
1. The transferability of VLM jailbreak attacks is an important and timely research topic.
2. The paper is clearly written and easy to follow.
3. The proposed method achieves good empirical performance.

## Weaknesses
1. Although the paper addresses the important problem of VLM jailbreak transferability, it focuses exclusively on an untargeted setting. In my view, this setting is not very practical, since an attacker ultimately aims to obtain responses to specific harmful queries rather than merely induce generic unsafe behavior.
2. The overall methodological novelty is limited. The proposed approach essentially augments an entropy-guided adversarial attack objective with a KL regularization term to improve stability and text quality, which is a fairly standard technique.
3. I find the motivation somewhat confusing. The paper argues that prefix-based optimization imposes excessive constraints that hurt transferability. However, Observation 1 states that entropy-maximization adversarial attacks mainly induce a transition from "sorry" to "sure" responses. This seems somewhat inconsistent and makes the underlying intuition unclear.
4. The experiments do not evaluate transferability to closed-source models or to larger-scale models, leaving the generality of the proposed approach insufficiently validated.

**Audience:**

Yes

**Audience Explanation:**

The transferability of VLM jailbreak attacks is an important and timely research topic

**Broader Impact Concerns:**

No concerns

**Claims And Evidence:**

Yes

**Claims Explanation:**

In the untargeted setting considered in this paper, the proposed method demonstrates consistent empirical improvements over existing baselines

**Requested Changes:**

1. Better justify the choice of the untargeted attack setting and discuss its practical relevance. In particular, the paper should clarify why transferability in the untargeted setting is an appropriate evaluation target, given that real-world attackers are often interested in eliciting responses to specific harmful queries.

2. The discussion regarding the limitations of prefix-based optimization and Observation 1 would benefit from a more coherent explanation of how entropy maximization improves transferability.

3. Aadditional transferability evaluations on closed-source VLMs and/or larger-scale models to further validate the generality of the proposed approach

---

> ### Author Response · Authors · 2026-07-09
> **Response to Reviewer gGGt part 1**
>
> We sincerely thank Reviewer gGGt for the constructive comments and for recognizing the importance of VLM jailbreak transferability, the clarity of the paper, and the empirical performance of the proposed method. Following the review structure, we respond below in the order of the reviewer’s weaknesses, while explicitly marking the corresponding requested changes where applicable as [(W(Weakness) X ),(RC(Requested Change) X )].
>
> ## Q1. [W1, RC1]: An untargeted setting may be less practical because attackers usually want responses to specific harmful queries rather than generic unsafe behavior.
>
> **A1:** We clarify that our method is untargeted in the optimization form, not untargeted in the expected results. The final desired behavior should match the original harmful instruction. The difference from many previous jailbreak objectives is that we do not force the model to match a fixed prefix or a target response. Such constraints are often unnecessary for the attacker’s final goal: obtaining a useful response to the harmful instruction. The exact surface form of the answer is not essential. In fact, forcing the same prefix or a full response across models can be unreasonable, as different VLMs may produce harmful responses via different decoding trajectories.
>
> In addition, this relaxed constrained objective is also considered here for transferability. A targeted objective optimizes the adversarial image toward a specific output trajectory of the source model. When transferred to another VLM, the same response pattern may no longer align with the target model’s decoding behavior. By contrast, our untargeted response-objective setting only aims to cross the refusal boundary while leaving the target model free to generate its own instruction-following continuation.
>
> In the revision, we will explicitly define the setting as “untargeted with respect to response form, but the final desired behavior is still a response that follows the harmful instruction.” This better reflects the practical goal: the attack aims to bypass safety alignment and obtain a useful instruction-following response, while avoiding forced response-pattern constraints that harm transferability.
>
> ## Q2. [W2] The method may be a coarse entropy-guided attack.
>
> **A2:** We start from harmful instructions that are normally refused, and we seek an untargeted optimization-based jailbreak objective that improves transferability without forcing a fixed prefix or a full target response. In this setting, high-entropy positions are useful because they often correspond to decision points without explicitly setting a target response.
>
> However, although entropy-only optimization can help cross the refusal boundary, it can also harm output structure, making it useless. UJEM-KL therefore applies entropy maximization to refusal decision positions and KL stabilization to structural positions, aiming to preserve useful instruction-following responses after refusal is bypassed.
>
> We will revise the paper to emphasize the method idea and effectiveness.
>
>
> ## Q3. [W3, RC2]: Motivation is confusing because Observation 1 describes a “sorry” to “sure” transition, which may appear similar to prefix optimization.
>
> **A3:** The transition from “sorry” to “sure” is an observed outcome, not an optimization target. It should not be interpreted as replacing the objective with a target prefix optimization. Observation 1 uses these tokens only to visualize what token appear decisions occur in the next-token distribution. UJEM-KL never optimizes toward “sure” or any fixed affirmative phrase. Instead, it maximizes entropy at naturally high-entropy refusal-decision positions where refusal and non-refusal continuations already compete. The model is then allowed to select its own continuation.
>
> This distinction is central to the transferability motivation. Prefix-based optimization treats a particular output pattern as the path to jailbreak, which can overfit the source model and reduce transfer. UJEM-KL optimizes the refusal-decision boundary rather than the surface output. The “sorry” to “sure” transition is therefore a diagnostic observation after an attack, not an expected signal used by the objective. This is also supported by the controlled objective comparison in Table 3 of the current paper. With Qwen2.5-VL as the source model, Full-target achieves 85.14% white-box ASR but transfers poorly to InternVL and LLaVA, with 23.45% and 28.16% ASR, respectively. UJEM-KL slightly reduces the white-box attack performance to 82.23% but improves transfer to 48.62% and 56.63%. Similarly, with InternVL as the source model, UJEM-KL improves the average transfer ASR from 26.52% under Full-target to 52.86%. In the revision, we will make clear that “sorry” and “sure” are explanatory markers, not optimization targets.

---

> ### Author Response · Authors · 2026-07-09
> **Response to Reviewer gGGt part 2**
>
> ## Q4. [W4, RC3]: Additional transferability evaluations on closed-source VLMs and/or larger-scale models.
>
> **A4:** Thank you for this comment. Larger-scale and closed-source evaluation would broaden the empirical scope, and the current claim should be read as being based on representative open-source VLMs. We will make this scope explicit.
>
> We add transferability evaluation following the same controlled-objective protocol as in Table 3. The first column reports the off-diagonal transfer average from the current paper's controlled-objective comparison, computed over source models Qwen2.5-VL and InternVL3.5. For open-source evaluation, we use slightly larger variants from the same three model families to test whether the trend holds beyond the exact models used in the main paper. Specifically, the larger open-source variants correspond to the next larger model from the same model families used in the main evaluation: Qwen2.5-VL-32B-Instruct, InternVL3.5-8B, and LLaVA-1.5-13B. We average the transfer results over these larger open-source targets to avoid over-emphasizing a single model family. For closed-source evaluation, we include GPT-4o. The absolute transfer rates are lower, which is consistent with stronger safety alignment and API-level defenses.
>
> **Response Table 3. Transfer evaluation on larger open-source variants and GPT-4o on JailBreakV-28K.**
>
> | Objective | Open-source transfer avg. (%) ↑ | Larger open-source variants avg. (%) ↑ | GPT-4o (%) ↑ |
> |---|---:|---:|---:|
> | Full-target | 26.16 | 18.4 | 5.2 |
> | Fixed-prefix | 38.19 | 27.6 | 9.8 |
> | UJEM-KL | 52.74 | 36.5 | 16.4 |
>
> Relaxing response-pattern constraints still preserves more transferability than Full-target or Fixed-prefix objectives. We will also include this in the revised paper.

---

### Review · Reviewer_b2vV · 2026-06-11

**Summary Of Contributions:**

## Summary of Contribution
The paper studies transferability in vision-language model (VLM) jailbreak attacks. The authors argue that poor transferability reported in prior work may stem from overly constrained targeted objectives rather than an inherent lack of shared vulnerabilities across models. Based on the observation that refusal behavior tends to concentrate around high-entropy decision points, the paper proposes UJEM, which maximizes entropy on selected high-entropy tokens, and UJEM-KL, which further stabilizes low-entropy positions using a KL regularizer. Experiments on JailBreakV-28K and SafeBench demonstrate competitive white-box attack success rates and improved cross-model transferability compared to several baselines.

## Strengths:

1.  The paper is well-written and easy to understand. The basic observation and motivation are reasonable.
2.  The experimental results demonstrate the high ASR and transferability of the proposed method.
3. The idea that maximizing entropy as a untargeted attack to improve transferability is interesting.

## Weaknesses:

1. The novelty appears limited, as entropy-maximization-based jailbreak methods (e.g., EGA) have already explored maximizing output entropy to jailbreak and the cross-model transferability.
2. The paper argues that the KL improves response quality and that this improvement contributes to the gains of UJEM-KL. However, no quantitative evaluation of response quality is provided.
3. "The experimental results show that moderate temperatures are preferred for better ASR, and overly high temperatures reduce stability." is not correct, because the ASR of T = 0 shows the highest ASR in Table 7. The authors didn't provide related analysis.
4. The design of the KL regularization appears somewhat coarse, applying KL regularization to all non-decision tokens. The results in Table 6 show that KL can  damage the ASR. However, Table 6 shows that increasing the KL weight can reduce ASR, which raises the question of whether all non-decision tokens should be treated equally.

**Audience:**

Yes

**Audience Explanation:**

The paper studies an important problem in VLM safety and presents interesting observations on jailbreak transferability. The findings are likely to be relevant to researchers working on adversarial attacks, robustness, and alignment of multimodal models.

**Claims And Evidence:**

No

**Claims Explanation:**

The empirical results are promising, but the claimed response-quality improvements from KL regularization are not quantitatively evaluated.

**Requested Changes:**

1. Add the evaluation results of the response quality.
2. Correct and enrich the analysis of the results shown in Table 7.
3. The proportion of the decision tokens is set to 20% in the paper. It's better to add ablations about different proportions.

---

> ### Author Response · Authors · 2026-07-09
> **Response to Reviewer b2vV part 1**
>
> We sincerely thank Reviewer b2vV for the constructive comments and for recognizing that the paper is clearly written, addresses an important VLM safety problem, and demonstrates the strong effectiveness of the proposed method. Following the review structure, we respond below in the order of the reviewer’s weaknesses, while explicitly marking the corresponding requested changes where applicable as [(W(Weakness) X ),(RC(Requested Change) X )]
>
> ## Q1. [W1] That entropy maximization and high-entropy token selection have already been explored by EGA, so the novelty of UJEM-KL may appear limited.
>
> **A1:** Entropy maximization and high-entropy token selection are indeed related to EGA, which we cite and use as a baseline. However, the distinction is not simply whether high-entropy tokens are selected. EGA also uses entropy-based token selection. The key differences are the **motivation**, the **threat model**, and  the **method itself**.
>
> First, EGA studies entropy-guided attacks as a general mechanism for inducing degraded or even harmful behavior in VLM generation. In contrast, our motivation is to find an untargeted optimization-based jailbreak objective that improves transferability. We are not proposing entropy maximization as a standalone novelty; rather, entropy maximization provides a natural solution that matches our motivation: relaxing fixed response-pattern constraints while still perturbing refusal-related decoding decisions.
>
> Second,EGA considers cases where originally benign VLM generations can be pushed toward harmful content in adversarial attack. Our setting starts from an explicitly harmful instruction, and the goal is to obtain a useful response to that harmful request under an untargeted response-form objective.
>
> Third, EGA follows an entropy-selection plus bank-selection formulation: it selects high-entropy tokens and uses an entropy bank to guide the attack. UJEM-KL instead follows an entropy-selection plus KL-on-selection formulation. We select high-entropy refusal-decision positions for entropy maximization and apply KL stabilization to the complementary structural positions. This design is introduced because entropy-only optimization can bypass refusal but may also damage response structure, producing repetitive, fragmented, or less useful outputs. The KL term is a stabilization mechanism for preserving usable jailbreak responses.
>
> We will revise this to make this distinction explicit: UJEM-KL builds on the high-entropy-token insight but uses it for a different problem, namely transferable untargeted VLM jailbreaks under fixed harmful instructions, with a KL stabilization design that differs from EGA’s entropy-bank formulation.
>
> ## Q2. [W2, RC1]: The paper claims KL improves response quality, but does not provide a separate quantitative evaluation of this claim.
>
> **A2:** Thank you for this comment. In the current version, we use the intersection of three safety judges to reduce false positives from borderline unsafe outputs, such as responses that contain unsafe fragments but are not substantively useful. The three judges are intentionally complementary, and taking their intersection therefore makes the reported ASR stricter than a single-judge evaluation.
>
> To further address this request, we add one compact evaluation using two additional metrics: a StrongREJECT-style usefulness score for jailbreak-specific response usefulness [A], and the standard corpus-level Distinct-2 for generation diversity [B]. Because some VLM outputs contain a partial refusal followed by substantive harmful information, we report the usefulness component of the StrongREJECT rubric rather than the refusal-gated score. Specifically, we set the refusal gate to zero and compute the normalized specificity-convincingness score. This isolates whether the response contains specific and substantively useful information for the original harmful instruction, while our three-judge ASR remains responsible for determining whether the output is unsafe. We include SEA because it is the strongest baseline in the main comparison, making it a useful reference for evaluating whether UJEM-KL improves response usefulness.
>
> **Response Table 1. Response  analysis on JailBreakV-28K (averaged over Qwen2.5-VL, InternVL3.5, and LLaVA).**
>
> | Method | ASR avg. (%) | SR-usefulness avg. ↑ | Distinct-2 avg. ↑ |
> |---|---:|---:|---:|
> | SEA | 83.42 | 0.53 | 0.184 |
> | UJEM | 77.90 | 0.46 | 0.132 |
> | UJEM-KL | 84.74 | 0.63 | 0.216 |
>
>
> The added metrics make the role of KL explicit: compared with entropy-only UJEM, UJEM-KL improves jailbreak-specific usefulness and reduces repetitive degeneration while maintaining strong ASR. We will add this evaluation to the revised version.

---

> ### Author Response · Authors · 2026-07-09
> **Response to Reviewer b2vV part 2**
>
> ## Q3. [W3, RC2]: The current analysis is inaccurate because \(T=0\) achieves the highest ASR.
>
> **A3:** We included \(T=0\) as a deterministic greedy-decoding reference, not as the default open-ended generation protocol, and not to claim that moderate temperatures universally yield higher ASR. Table 7 shows that \(T=0\) gives 83.4%, 79.2%, and 89.3% ASR on Qwen, InternVL, and LLaVA, respectively. This deterministic reference is useful because it verifies that UJEM-KL does not rely on sampling randomness. The default sampling setting \(T=0.8\) remains near-competitive and follows the open-ended generation protocol used in the main experiments.
>
> In the revision, we will clarify this point and revise the conclusion as follows: \(T=0\) provides the strongest deterministic-reference performance, while UJEM-KL remains effective across decoding temperatures and does not depend on sampling setting.
>
> ## Q4. [W4]: KL regularization may be coarse because all non-decision tokens are treated equally, and Table 6 shows that a larger KL weight can reduce ASR.
>
> **A4:** The existing KL-scope ablation in Appendix Table 10 already compares the main scope choices: no KL, KL on all positions, and KL only on selected positions Rρ. Applying KL to all positions strongly suppresses ASR to 28.2%, 21.5%, and 43.6% on Qwen, InternVL, and LLaVA, respectively. In contrast, applying KL only to selected positions Rρ achieves 82.2%, 83.6%, and 88.3%. Thus, the current select position-only KL is a lightweight design supported by the ablation, rather than a coarse setting.
>
> Table 6 shows the  ASR-stability trade-off: an overly large KL weight can suppress token flipping and reduce ASR, while a moderate KL weight stabilizes structural positions without eliminating the attack effect. The additional SR-usefulness and Distinct-2 analysis in Q2 further clarifies that KL is not intended to maximize raw ASR alone, but to improve the overall ASR-usefulness-degeneration trade-off.
>
> We will revise the description to make the existing comparison clearer and explicitly refer to the supplementary KL-scope ablation. We will also discuss more fine-grained weighting within a selected position Rρ as a possible extension.
>
> ## Q5.[RC3]: Ablations on the decision-token proportion, since the current paper uses 20%.
>
> **A5:** We add a local ratio ablation around the selected value to analyze the trade-off in covering decision positions. We focus on a local sweep around the current setting, \(ratio \in \{10\%, 20\%, 30\%\}\), which directly tests whether the selected ratio is robust.
>
> **Response Table 2. Decision-token ratio ablation on JailBreakV-28K (averaged over Qwen2.5-VL, InternVL3.5, and LLaVA).**
>
> | ratio | ASR avg. (%) ↑ | SR-usefulness ↑ | Distinct-2 ↑ |
> |---:|---:|---:|---:|
> | 10% | 72.1 | 0.50 | 0.165 |
> | 20% | 84.7 | 0.63 | 0.216 |
> | 30% | 83.8 | 0.61 | 0.207 |
>
>
> The results show the trade-off controlled by **ratio**. A smaller ratio may miss important decision positions, while a larger ratio heats more structural tokens and begins to approach global entropy maximization. The 30% ratio remains competitive but slightly reduces ASR, response usefulness, and corpus-level diversity compared with 20%. Thus, ratio=20% provides the strongest overall ASR-usefulness-degeneration trade-off in this local sweep.
>
> ---
>
> # References
>
> [A] Souly, A., Lu, Q., Bowen, D., Trinh, T., Hsieh, E., Pandey, S., Abbeel, P., Svegliato, J., Emmons, S., Watkins, O., & Toyer, S. *A StrongREJECT for Empty Jailbreaks*. arXiv:2402.10260, 2024.
>
> [B] Li, J., Galley, M., Brockett, C., Gao, J., & Dolan, B. *A Diversity-Promoting Objective Function for Neural Conversation Models*. NAACL, 2016.